# The Protective Effect of New Carnosine-Hyaluronic Acid Conjugate on the Inflammation and Cartilage Degradation in the Experimental Model of Osteoarthritis

Rosalba Siracusa [1], Daniela Impellizzeri [1], Marika Cordaro [1], Alessio F. Peritore [1], Enrico Gugliandolo [1], Ramona D'Amico [1], Roberta Fusco [1], Rosalia Crupi [1], Enrico Rizzarelli [2], Salvatore Cuzzocrea [1,3,*], Susanna Vaccaro [4], Mariafiorenza Pulicetta [4], Valentina Greco [2], Sebastiano Sciuto [2], Antonella Schiavinato [5], Luciano Messina [4] and Rosanna Di Paola [1]

[1]  Department of Chemical, Biological, Pharmaceutical and Environmental Science, University of Messina, 98166 Messina, Italy; rsiracusa@unime.it (R.S.); dimpellizzeri@unime.it (D.I.); cordarom@unime.it (M.C.); peritoreal@gmail.com (A.F.P.); egugliandolo@unime.it (E.G.); rdamico@unime.it (R.D.); rfusco@unime.it (R.F.); rcrupi@unime.it (R.C.); dipaolar@unime.it (R.D.P.)
[2]  Department of Chemical Sciences, University of Catania, Viale A. Doria 6, 95125 Catania, Italy; erizzarelli@unict.it (E.R.); vgreco@unict.it (V.G.); ssciuto@unict.it (S.S.)
[3]  Department of Pharmacological and Physiological Science, Saint Louis University School of Medicine, Saint Louis, MO 63104, USA
[4]  Fidia Farmaceutici, 96017 Noto (SR), Italy; svaccaro@fidiapharma.it (S.V.); mpulicetta@fidiapharma.it (M.P.); LMessina@fidiapharma.it (L.M.)
[5]  Fidia Farmaceutici, 35031 Abano Terme (PD), Italy; ASchiavinato@fidiapharma.it
*  Correspondence: salvator@unime.it; Tel.: +39-090-6765208

**Abstract:** Osteoarthritis (OA) is a disease that currently has no cure. There are numerous studies showing that carnosine and hyaluronic acid (HA) have a positive pharmacological action during joint inflammation. For this reason, the goal of this research was to discover the protective effect of a new carnosine conjugate with hyaluronic acid (FidHycarn) on the inflammatory response and on the cartilage degradation in an in vivo experimental model of OA. This model was induced by a single intra-articular (i.ar.) injection of 25 μL of normal saline with 1 mg of monosodium iodoacetate solution (MIA) in the knee joint of rats. MIA injection caused histological alterations and degradation of the cartilage, as well as behavioral changes. Oral treatment with FidHycarn ameliorated the macroscopic signs, improved thermal hyperalgesia and the weight distribution of the hind paw, and decreased histological and radiographic alterations. The oxidative damage was analyzed by evaluating the levels of nitrotyrosine and inducible nitric oxide synthase (iNOS) that were significantly reduced in FidHycarn rats. Moreover, the levels of pro-inflammatory cytokines and chemokines were also significantly reduced by FidHycarn. Therefore, for the first time, the effectiveness of oral administration of FidHycarn has been demonstrated in an osteoarthritis model. In conclusion, the new FidHycarn could represent an interesting therapeutic strategy to combat osteoarthritis.

**Keywords:** osteoarthritis; carnosine; hyaluronic acid; inflammation; oxidative stress

## 1. Introduction

Osteoarthritis (OA) is one of the most common arthropathies and is the leading cause of disability, with a large socioeconomic cost. OA is a condition which can affect any joint in the body. OA generally

affects the joints that support most of the weight (such as the knees and feet) and the joints we use the most (e.g., hand joints). In a healthy joint, the cartilage shelters the surface of the bones and supports the bones to move liberally against each other. When a joint develops OA, part of the cartilage tapers and the surface becomes coarser. This means the articulation does not move as easily as it should. When the cartilage becomes worn out or damaged, all the tissues within the articulation become more active than usual as the body attempts to repair the damage. However, the reparation processes do not always operate well, and thus, modifications to the joint structure can occasionally cause or contribute to symptoms such as swelling, pain, and ultimately, disability.

One of the leading characteristics of OA is collagen deterioration, as suggested by augmented tissue swelling and the loss of proteoglycans [1]. Both collagen destruction and the loss of proteoglycans unfavorably affect the mechanical properties of cartilage. Chondrocytes respond to tissue injury by increasing collagen and proteoglycan synthesis in an attempt to repair the tissue [2]. If repair fails, the damage will progress to articular cartilage degeneration. It is still not clear exactly what causes osteoarthritis. We do know that it is not simply 'wear and tear' and that the danger of developing OA depends on a number of factors.

Furthermore, these degradation processes are thought to be largely elicited through excess production of pro-inflammatory and catabolic mediators. Among them, interleukin-1β (IL-1β) has been demonstrated to be predominantly involved in both disease initiation and progression [3,4]. Oxidative stress shows a fundamental role in supporting cartilage degradation. In chondrocytes of the joint, the data implicate reactive oxygen species (ROS) as signaling intermediates of IL-1β [5]. It has been proposed that ROS produced internally in the joints may contribute notably to the pathogenesis of OA, as these inorganic oxidants are able to deteriorate cartilage through oxidation of the extra-cellular matrix (ECM) components or post-translational modification (PTM) of metalloproteinases (MMPs) [6,7].

The existing treatments for postponing OA and rheumatoid arthritis (RA) progression comprise of some disease modifying anti-rheumatic drugs (DMARDs) and biological agents that operate as immunomodulatory drugs in RA [8]. Several also act by inhibiting endothelial cell proliferation and cytokines [9]. Moreover, all of these compounds have potentially grave side effects and there are substantial differences in toxicity among DMARDs [10]. Hyaluronan (HA) is a foremost component of synovial fluid (SF) and is necessary for the proper functioning of joints. Yet, owing to the fact that SF does not have any hyaluronidases, it is considered that ROS could also be involved in HA catabolism in the inflamed articulation. A transition metal, e.g., copper or iron, would also be required to show an active function in oxidative HA catabolism. The polymer functions of HA are dimension specific and its fragments represent an information-rich system. The intermediate-sized HA-polymer fragments are inflammatory, immune-stimulatory, and highly angiogenic [11].

Consequently, substances preventing HA from being degraded could have anti-inflammatory and anti-angiogenic properties. Carnosine (CARN), a fundamental endogenous molecule, has several physiological functions: pH buffering, radical scavenging, anti-glycating, heavy metal chelating, and neutralization of toxic aldehydes. CARN is found to have neuroprotective, cataract treating, hepatoprotective, and anti-aging abilities [12]; however, its anti-inflammatory strength in auto-immune systemic inflammatory diseases has been recently investigated. The ambition of this study is to explore the effect of FidHycarn on the inflammatory response and on the cartilage degradation in the experimental rat model of OA.

## 2. Material and Methods

### 2.1. Animals

Sprague-Dawley male rats (200–230 g, 7-weeks old, Envigo, Udine, Italy), were used for this study. Ten rats were used for each treatment group (see below). Food and water were available ad libitum. This study was conducted in accordance with the University of Messina Review Board for the care and

use of animals. Animal care was in accordance with the Italian regulations on safety of animals used and other scientific purposes (D.Lgs 2014/26 and EU Directive 2010/63).

## 2.2. Osteoarthritis Induction and Treatment

OA was induced by intra-articular (i.ar.) injection of monosodium iodoacetate solution (MIA) in the knee joint of rats, as previously described [13]. On the first day, the rats were anesthetized and only one injection of 25 µL germ-free normal saline containing 1 mg MIA was injected into the right knee articulation through the infra-patellar ligament. The left knee received an identical volume of 0.9% saline. MIA and drug solutions were prepared under germ-free conditions and injected with a 50 µL Hamilton microliter syringe with a 6-mm, 27-gauge needle that was introduced into the joint circa 2–3 mm.

## 2.3. Experimental Groups

Rats were distributed into the following experimental groups:

**Sham—Control:** The rats were exposed to intra-articular injection with 0.9% saline (25 µL) instead of MIA and were orally treated with vehicle (distilled water) every day starting from day 3 after MIA induction; (n = 20).

**MIA—Vehicle:** The rats were subjected to MIA (as indicated below) (1 mg/kg) and were orally treated with distilled water (vehicle for FidHycarn) every day starting from day 3 after MIA induction; (n = 20).

**MIA—Carnosine:** The rats were subjected to MIA (1 mg/kg) and received oral Carnosine (17.6 mg/kg) every day starting from day 3 after MIA induction; (n = 20).

**MIA—HA:** The rats were subjected to MIA (1 mg/kg) and received oral HA (70.9 mg/kg) every day starting from day 3 after MIA induction; (n = 20).

**MIA—Carnosine+HA:** The rats were subjected to MIA (1 mg/kg) and received oral Carnosine+HA (17.6 + 70.9 mg/kg) every day starting from day 3 after MIA induction; (n = 20).

**MIA—FidHycarn (carnosine conjugate with hyaluronic acid):** The rats were subjected to MIA (1 mg/kg) and received oral FidHycarn (88.5 mg/kg) every day starting from day 3 after MIA induction; (n = 20).

**MIA—Naproxen:** The rats were subjected to MIA (1 mg/kg) and treated orally with Naproxen (10 mg/kg) every day starting from day 3 after MIA induction. (n = 20).

The beginning of treatment was 3 days after MIA injection to allow the development of the first cartilage damages [14].

## 2.4. Synthesis of FidHycarn

The synthesis of FidHycarn is described in the patent WO2019069258 [15]. Briefly, the first step is the synthesis of carnosine methyl ester. We treated 1.5 g of carnosine with 50 mL of an acetyl chloride solution in anhydrous methanol (pre-mixed) in a 1:20 ratio (*v/v*), under stirring inside a 250 mL flask, and successively, about 90% of the solvent was removed by evaporation. 20 mL of anhydrous methanol was then added to the reaction residue, and again, 90% of the solvent was removed by evaporation. The operation was repeated until all HCL (which was formed during the reaction) had been removed; the product was then brought to dryness under a vacuum. Following this step, 1.1 g of about 700 kDa hyaluronic acid were introduced in a reactor with 80 mL of a mixture of $H_2O$ and DMSO. Successively, a solution of $H_2O$ and DMSO containing tris 2-(2 methoxethoxy) ethyl amine, 3-hydroxy 1,2,3 benzotriazin 4 (**3H**) one, and *N* (3 dimethylaminopropyl) *N* ethylcarbodiimide hydro-chloride was added. Successively, 365 mg of carnosine methyl ester in DMSO were added. The product was precipitated with the addition of ethanol. The final precipitate was dissolved in water and subsequently lyophilized. An amount of 100 mg of the FidHycarn batch used in this in vivo study was constituted of 19.89 mg of carnosine and 80.11 mg of hyaluronic acid. In Figure 1 we represent the

new carnosine-hyaluronic acid conjugate. All FidHycarn samples were subjected to accurate [1]H-NMR analysis to define their structure and the % of carnosine loading (Figure 2).

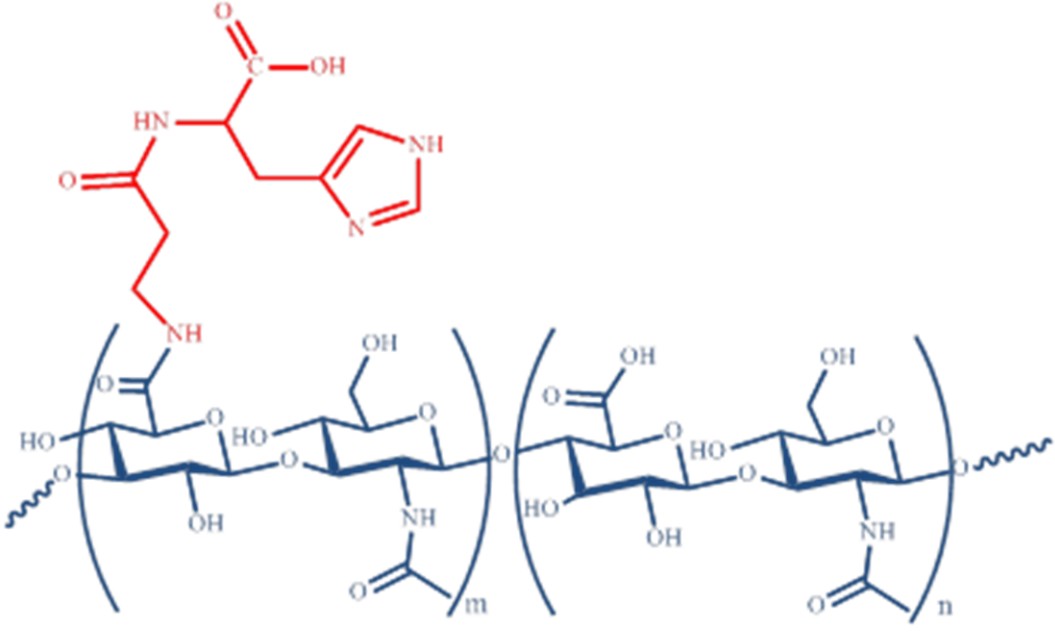

**Figure 1.** Carnosine-hyaluronic acid conjugate.

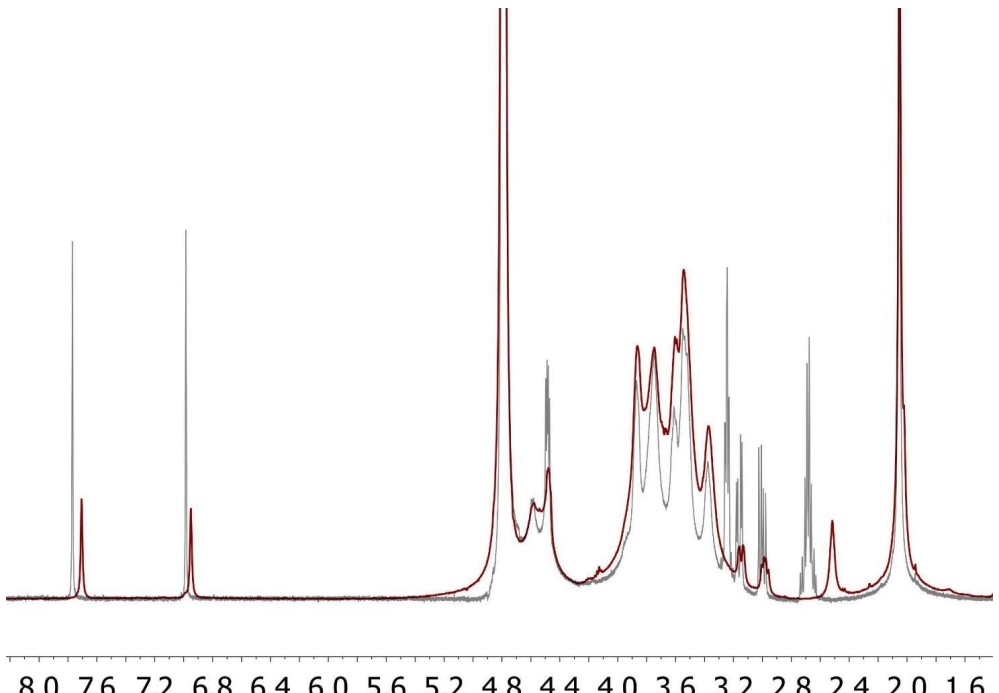

**Figure 2.** Overlapped [1]H-NMR spectra of carnosine in mixture with hyaluronic (CAR+HA; grey line) and of the carnosine-hyaluronic acid conjugate (red line). The signals attributable to the protons on C-2 of the beta-alanine moiety in the mixture (2.68 ppm) appear upfield shifted at 2.44 ppm in the spectrum of the conjugate (according to the transformation of the amino group into the relevant amide function). At the same time, the signals of protons on C-3 of beta-alanine in the mixture (3.2 ppm) appear downfield at 3.68 ppm in the spectrum of the conjugate (these are hidden by the signals of the hyaluronic acid backbone). The signal assignments were confirmed by bi-dimensional [1]H-[1]H-NMR (COSY experiments). Abbreviations: HA, hyaluronic acid.

### 3. Behavioral Analysis

The hyperalgesic reactions to heat were determined at different time points (days 0, 7, 10, 13, 16, and 21) using a Basile Plantar Test. Each rat was limited to staying in a plexiglass chamber and was allowed to habituate. A portable unit consisting of a high intensity projector lamp was placed to deliver a thermal stimulation directly to a hind paw from under the chamber. The retraction latency period of injected paws was defined with an electronic clock circuit and thermocouple. Results were expressed as paw-withdrawal latency (s) [16].

#### 3.1. Evaluation of Pain-Related Behavior

Pain connected with OA was characterized by changes in the weight distribution on each hind paw [17]. The animals were positioned into a plexiglass chamber with each hind paw on the distinct force plate, and they were allowed to habituate. When still, the force used on the plate by each hind paw was annotated. A total of four readings was taken for each rat at each time point (days 0, 7, 10, 13, 16, and 21) and the mean was used for calculation. The percent weight distribution of the left (ipsilateral) hind paw was calculated by the following formula:

% weight distribution of left hind paw = left weight/(left weight + right weight) × 100.

#### 3.2. Macroscopic Examination of the MIA Induced OA Site

The rats were euthanized by giving excess anesthesia on the 21[th] day of post-MIA injection. The whole right knee joint of all the animals was explanted for macroscopic study. Soft tissues surrounding the right knee were removed to observe the clear features of the articular cartilage. Macroscopic investigation was performed to find the deformities of the joint capsules [18].

#### 3.3. Radiographic Analysis

Radiographic analysis was performed with an X-ray Bruker FX Pro instrument (Milan, Italy). The radiographs were assessed by an observer in a blinded mode and scored by means of a semi-quantitative grading scale as previously indicated [19]. Briefly, the articular space was scored by a scale 0–3 with 0 = regular, 1 = mild, 2 = partial, and 3 = whole loss of joint space. Subchondral bone sclerosis was indicated as 0 = normal, 1 = mild, 2 = moderate, and 3 = severe. Osteophyte formation was recorded as 0 = normal, 1 = osteophytes on the proximal tibia, 2 = osteophytes on the femoral condyle, and 3 = osteophytes present on both the tibial and femoral condyle.

#### 3.4. Histological Investigation

On the 21[st] day post MIA administration, the rats were killed by anesthetic overdose and the tissues were fixed by transcardiac perfusion with 4% paraformaldehyde solution. The tibiofemoral joints were taken and post-fixed in neutral buffered formalin (having 4% formaldehyde), were decalcified in EDTA, and were treated as previously described. Mid-coronal tissue sections (5 μm) were stained for valuations. Slices were stained with Hematoxylin and Eosin (H&E), and studied using light microscopy (Dialux 22 Leitz; Leica Microsystems SpA, Milan, Italy). The histopathological examination of the cartilage was assessed using the modified score of Mankin et al. (score range 0 to 12, from normal to complete disorganization and hypocellularity) [20]. Cartilage degeneration was assessed by staining with toluidine blue and analyzed using the following criteria described by Janusz et al.: 1 = mild into the surface region; 2 = slightly extended in the upper center; 3 = moderate in the median area; 4 = extended area deep; and 5 = severe degeneration [21].

#### 3.5. Immunohistochemistry

Briefly, the slides for immunohistochemistry (7 μm) were prepared from paraffin-embedded tissues. After deparaffinization, endogenous peroxidase was slaked with 0.3% hydrogen peroxide in 60%

methanol for 30 min. The slides were permeabilized with 0.1% Triton X-100 in PBS for 20 min. Sections were incubated overnight with anti-nitrotyrosine (1:500 in PBS, *w/v*; Millipore Sigma, Burlington, MA, USA, Cat. 06-284); anti-iNOS antibody (1:500 in PBS, *w/v*; BD Biosciences, San Jose, CA, USA, Cat. 610600); anti-IL-1β (1:500 in PBS, *v/v*; Santa Cruz Biotechnology, Heidelberg, Germany, Cat. sc-52012); and anti-interleukin (IL)-6 (1:500 in PBS, *w/v*; Santa Cruz Biotechnology, Heidelberg, Germany, Cat. sc-28343). Sections were washed with PBS, and then incubated with the secondary antibody. Specific labelling was identified with a biotin-conjugated goat anti-rabbit IgG and avidin-biotin peroxidase complex. The counter stain was developed with diaminobenzidine (brown) and nuclear fast red (red background). To verify that the immune-reaction for the nitrotyrosine is specific, we incubated some slices with the primary antibody in the presence of excess nitrotyrosine (10 mM). Similarly, to verify the binding specificity for iNOS, IL-1β, and IL-6, we incubated several slices with only the primary antibody (no secondary) or with only the secondary antibody (no primary).

As a general procedure, the digital images were opened in ImageJ, followed by deconvolution using the color deconvolution plug-in. When the IHC profiler plug-in is selected, it spontaneously plots a histogram profile of the deconvoluted DAB image, and a corresponding scoring log is demonstrated [22]. The histogram profile corresponds to the positive pixel intensity value obtained from the computer program [23].

### 3.6. Measurement of Cytokines

Tumor necrosis factor-α (TNF-α), interleukin (IL)-6, and IL-1β levels were evaluated in the plasma from MIA and sham rats. The assay was carried out using a colorimetric commercial ELISA kit (Calbiochem-Novabiochem Corporation, Milan, Italy).

### 3.7. Measurement of Chemokines

On the 21$^{st}$ day post MIA, the administration levels of chemokines MIP-1α and MIP-2 were estimated in the aqueous joint extracts. In brief, articular tissues were homogenized on ice in 3 mL of lysis buffer (PBS including 2 mM PMSF, and 0.1 mg/mL (final concentration), each of antipain, aprotinin, pepstatin A, and leupeptin) by means of a Polytron (Brinkinarm Instruments, Westbury, NY, USA). After homogenization, the tissues were centrifuged at 2000× *g* for 10 min. The supernatants were decontaminated with a Millipore filter (0.2 μm) and were conserved at −80 °C, if they are not analyzed immediately. The extracts generally contained 0.2 to 1.5 mg protein/mL, as quantified with a protein-assay kit (Pierce Chemical Co., Rockford, IL, USA). The levels of MIP-1α and MIP-2 have been quantified by means of a modification of a double-ligand method.

In brief, 96-well microliter plates were treated with 50 μL/well of rabbit anti-cytokine antibodies (1 μg/mL in 0.08 M NaOH, 0.26 $H_3BO_4$, and 0.6 M NaCl pH 9.6) for 16 h at 4 °C, and were then laved with wash buffer (containing PBS, 0.05% Tween-20, pH 7.5). Nonspecific binding sites on the plates were blocked with 2% BSA in PBS and were incubated for 90 min at 37 °C. Plates were cleaned four times with wash buffer, and diluted aqueous joint samples (50 μL) were added, followed by incubation for 1 h at 37 °C. After washing the plates, the chromogen substrate was added. The plates were incubated at room temperature to the wanted extinction, after which the reaction was stopped with 50 μL/well of 3 M $H_2SO_4$ solution. The plates were then read at 490 nm in an ELISA reader. This ELISA method consistently had a sensitivity limit of about 30 pg/mL.

## 4. Materials

All drugs were kindly offered from Fidia Farmaceutici S.p.A. Other compounds were acquired from Sigma-Aldrich Company (Milan, Italy). All substances were of the highest commercial grade available. All stock solutions were made in nonpyrogenic saline 0.9% NaCl (Baxter Healthcare Ltd., Thetford IP24 3SE, UK) or 10% ethanol (Sigma-Aldrich).

## 5. Data Evaluation

All values in the figures and text were expressed as standard error of the mean (SEM) of the mean of N observations. For the in vivo studies, N represents the number of animals studied. In the experiments concerning histology or immunohistochemistry, the figures shown are illustrative of at least three experiments performed on different experimental days on the tissue sections collected from all the animals in each group. Data sets were examined by one- or two-way analysis of variance followed by the Bonferroni test for multiple comparisons. A *p*-value of less than 0.05 was considered significant.

## 6. Results

### 6.1. Effects of FidHycarn on Thermal Hyperalgesia and Weight Distribution of the Hind Paw after OA Induction

Increased thermal hyperalgesia measured at different time points, as evidenced by a significant reduction in hind paw withdrawal latency as well as an important weight distribution asymmetry were observed in all MIA injected rats compared to the sham subjects. Oral treatments of carnosine at 17.6 mg/kg, carnosine+HA (17.6 + 70.9 mg/kg), and HA (70.9 mg/kg) were not able to ameliorate behavioral deficits. Instead, oral FidHycarn treatment at 88.5 mg/kg showed significant behavioral improvements (Figure 3A,B). No significant difference was found for hind paw withdrawal latency in the FidHycarn treatment group vs. the Naproxen treatment group, while the difference is significant for the weight distribution of the hind paw.

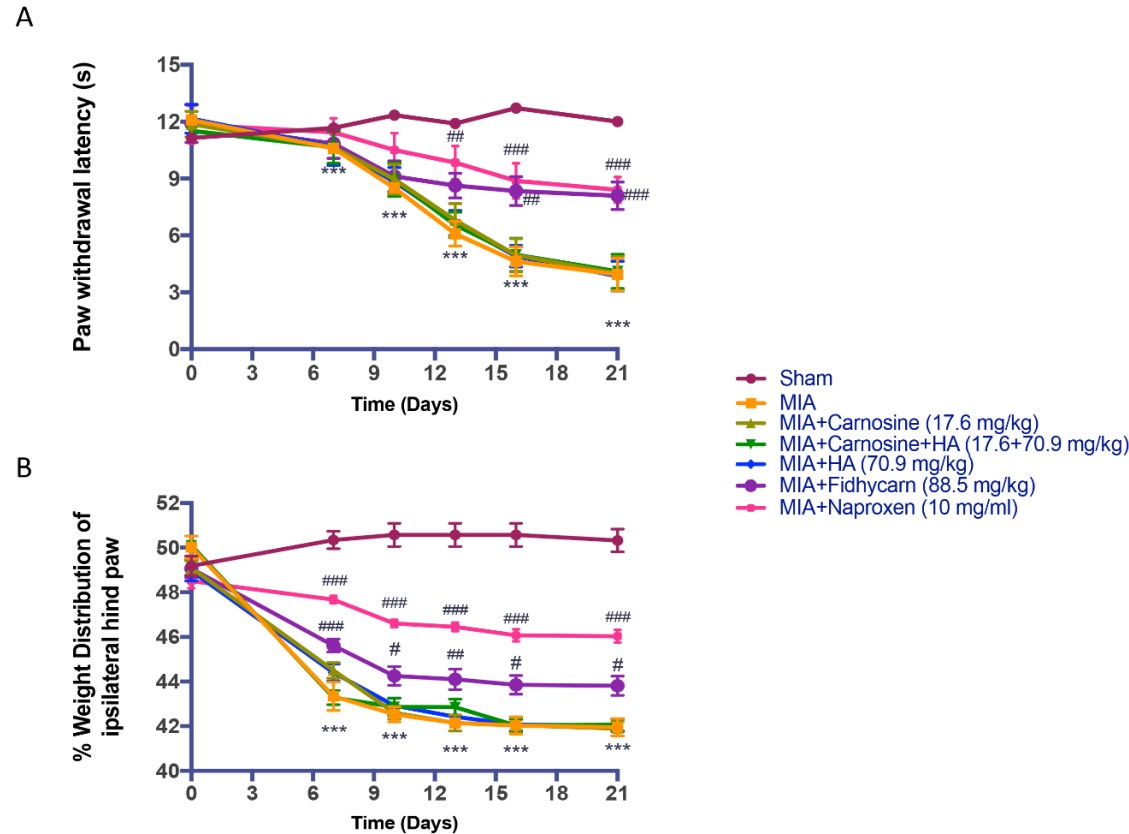

**Figure 3.** The effect of FidHycarn on behavior tests. Hyperalgesia was tested on the time-course indicated. Oral treatment with FidHycarn made a substantial improvement in hyperalgesia compared to other treatments (**A**). In addition, the treatment with FidHycarn showed a better distribution of the weight of the hind paw (**B**). *** $p < 0.001$ versus Sham; ### $p < 0.001$ versus MIA; ## $p < 0.01$ versus MIA; # $p < 0.05$ versus MIA. Abbreviations: MIA, monosodium iodoacetate solution.

## 6.2. Effects of FidHycarn on Macroscopic and Radiographic Analysis after MIA Injection

We found that 21 days after the i.ar. injection of MIA, the knee sections showed an important macroscopic and radiographic alteration in rats treated with the vehicle. Oral treatments of carnosine at 17.6 mg/kg, carnosine+HA (17.6 + 70.9 mg/kg), and HA (70.9 mg/kg) were not able to reduce these alterations. Instead, oral FidHycarn treatment at 88.5 mg/kg showed a substantial reduction of macroscopic and radiographic damage compared to the others. The effect of FidHycarn at 88.5 mg/kg was not statistically different compared to the Naproxen treatment. No damage was discovered in the sham group (Figure 4A,B and see radiographic analysis C).

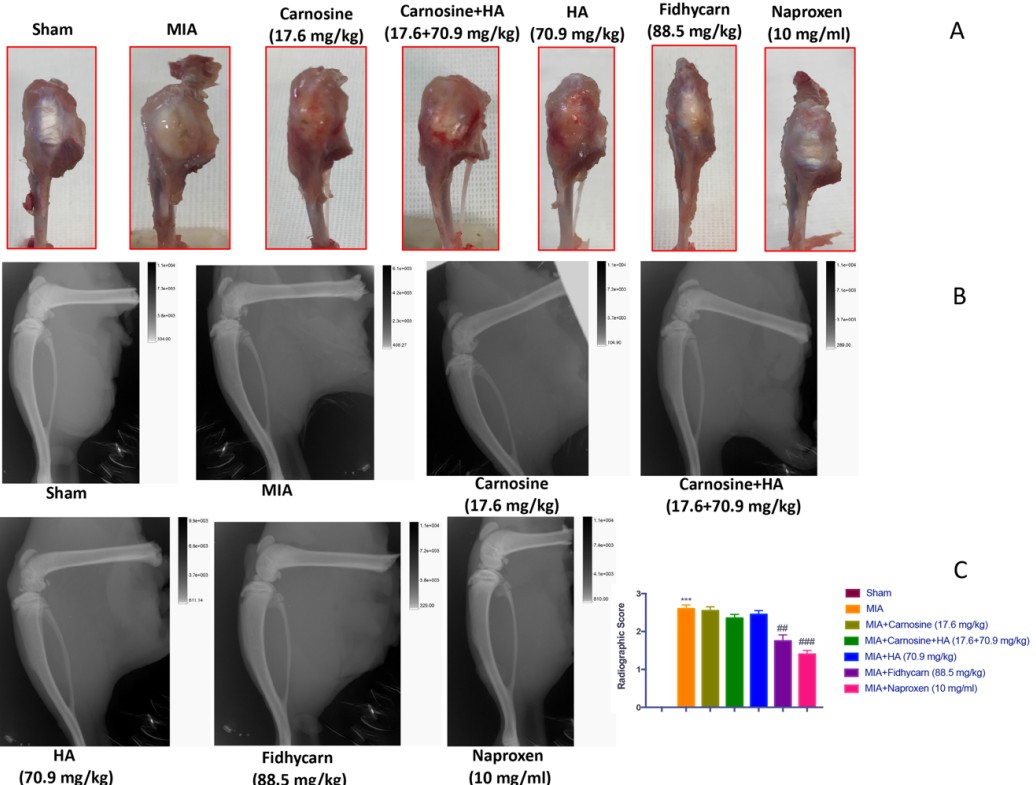

**Figure 4.** FidHycarn's macroscopic and radiographic effect on the knee joints. Macroscopic images for the thickening of the joint capsule of the animals of different groups (**A**). Oral treatment with FidHycarn displayed a significant improvement of the articular cartilage. This improvement was also observed by radiographic investigation where a diminution in the joint space was seen (**B**). Radiographic score (**C**). *** $p < 0.001$ versus Sham; ## $p < 0.01$ versus MIA; ### $p < 0.001$ versus MIA.

## 6.3. Effects of FidHycarn on Histological Damage and Cartilage Degeneration after MIA Injection

We stained the knee slices with H&E 21 days after the i.ar. injection of MIA. Histological investigation with light microscopy showed irregularities, disorganization, and fibrillation in the superficial layer and multilayering in the transition and radial zones, in the MIA injected rats in all experimental groups (Figure 5). In addition, cartilage degeneration was also evident by the blue color intensity of toluidine staining (Figure 6). Oral treatments of Carnosine at 17.6 mg/kg, Carnosine+HA (17.6 + 70.9 mg/kg), and HA (70.9 mg/kg) were not able to reduce this histological alteration and cartilage degeneration. Instead, oral FidHycarn treatment at 88.5 mg/kg showed an essential reduction of histological damage and cartilage degeneration compared to the others. The effect of FidHycarn at 88.5 mg/kg on cartilage degeneration was statistically different compared to Naproxen treatment but no statistical difference was observed for the histological damage. No injury was found in the Sham group.

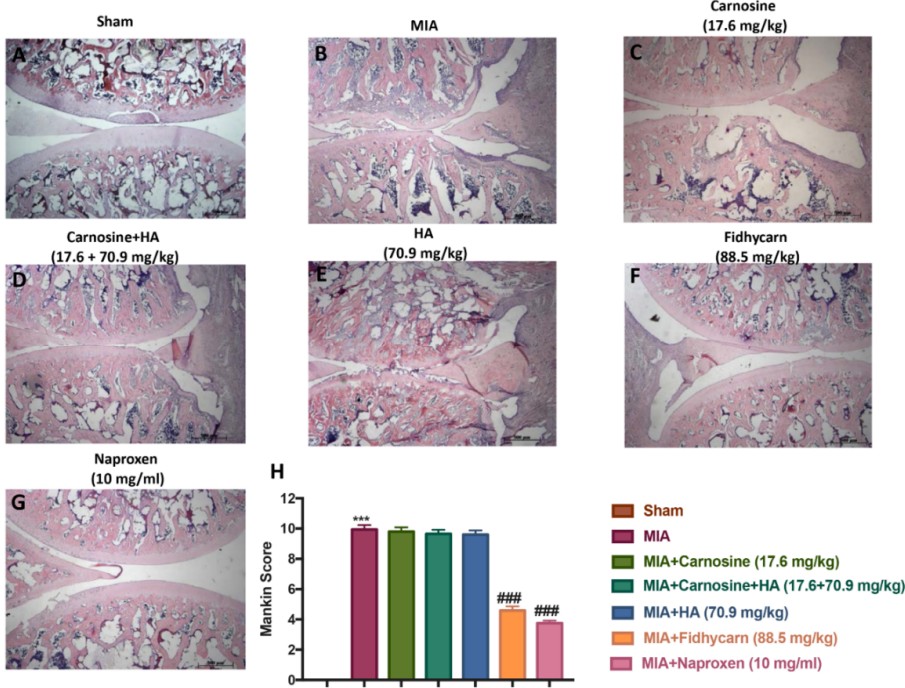

**Figure 5.** The effect of FidHycarn on the histological features of osteoarthritis (OA) knee tissue. The histological evaluation was performed by Hematoxylin and Eosin (H&E) staining. Panel (**A**), Sham; panel (**B**), MIA; panel (**C**), Carnosine; panel (**D**), Carnosine+HA; panel (**E**), HA; panel (**F**), FidHycarn; panel (**G**), Naproxen treatment. Figures are representative of all rats in each group. Panel (**H**), Mankin score for the treatment groups. *** $p < 0.001$ versus Sham; ### $p < 0.001$ versus MIA.

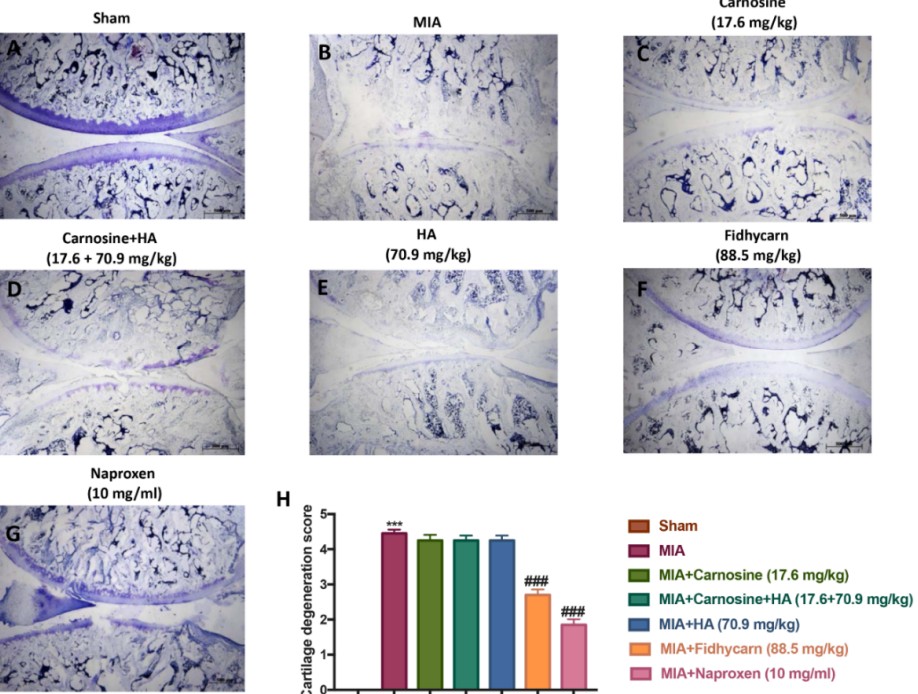

**Figure 6.** The effect of FidHycarn on cartilage degradation after MIA-induction. Cartilage evaluation by blue of toluidine-staining of joint sections showed significant degradation in MIA mice (**B**) compared to Sham animals (**A**). Ameliorated cartilage alterations were observed in the sections from MIA-Carnosine, MIA-Carnosine+HA, MIA-HA (**C–E**), and more significantly in FidHycarn treated mice (**F**). Naproxen is a positive control (**G**). The cartilage degradation score is shown in (**H**). Figures are representative of all animals in each group. *** $p < 0.001$ versus Sham; ### $p < 0.001$ versus MIA.

### 6.4. Effects of FidHycarn on Cytokine and Chemokine Levels after OA Induction

A substantial increase in TNF-α, IL-1β, IL-6, MIP-lα, and MIP-2 production was found in OA subjected rats 21 days post MIA injection. The levels of cytokines and chemokines were not significantly reduced in MIA rats treated orally with carnosine (17.6 mg/kg), carnosine+HA (17.6 + 70.9 mg/kg), and HA (70.9 mg/kg). On the contrary, oral FidHycarn treatment was able to reduce these levels. The effect of FidHycarn at 88.5 mg/kg on TNF-α, IL-6, and MIP-1α levels was statistically different to the naproxen treatment but no statistical difference was observed on IL-1β and MIP-2 levels. Low levels were found in the Sham animals (Figure 7).

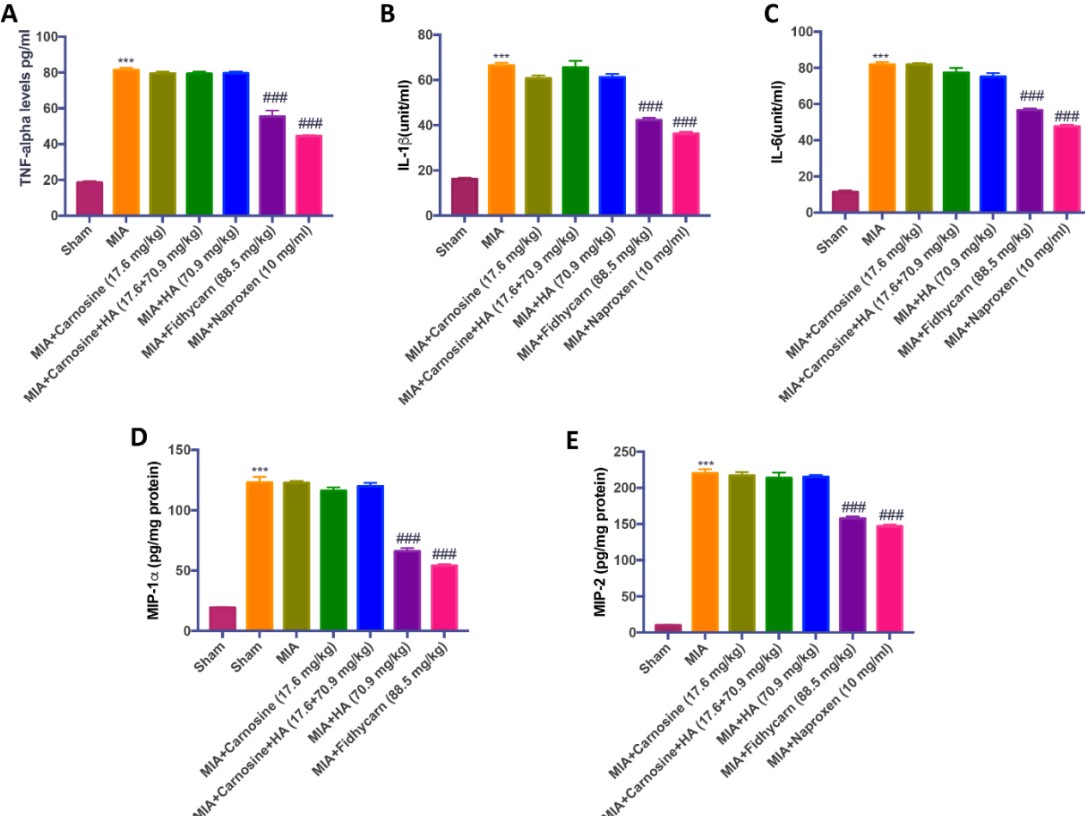

**Figure 7.** The effects of FidHycarn on cytokine and chemokine production. TNF-α (**A**), IL-1β (**B**), IL-6 (**C**), MIP-lα (**D**) and MIP-2 (**E**) levels were shown. Values are given as mean ± SEM of 20 animals for each group *** $p < 0.001$ versus Sham; ### $p < 0.001$ versus MIA.

### 6.5. The Effects of FidHycarn on Nitrotyrosine, INOS, IL-1β, and IL-6 Expression after OA Induction

A significant positive staining for nitrotyrosine, iNOS, IL-1β, and IL-6 was found in MIA rats treated with the vehicle (Figure 6, Figure 7, Figure 8, Figure 9B, and see H). Oral treatments of carnosine (17.6 mg/kg), carnosine+HA (17.6 + 70.9 mg/kg), and HA (70.9 mg/kg) were not able to reduce this positive staining (Figure 8, Figure 9, Figure 10, Figure 11C–E, and see H). Instead, oral FidHycarn treatment at 88.5 mg/kg showed an important reduction of positive staining for nitrotyrosine, iNOS, IL-1β, and IL-6 compared to the others (Figure 8, Figure 9, Figure 10, Figure 11F, and see H). The effect of FidHycarn at 88.5 mg/kg was statistically different to the Naproxen treatment (Figure 8, Figure 9, Figure 10, Figure 11G, and see H). No positive staining was discovered in the Sham group (Figure 8, Figure 9, Figure 10, Figure 11, and see H).

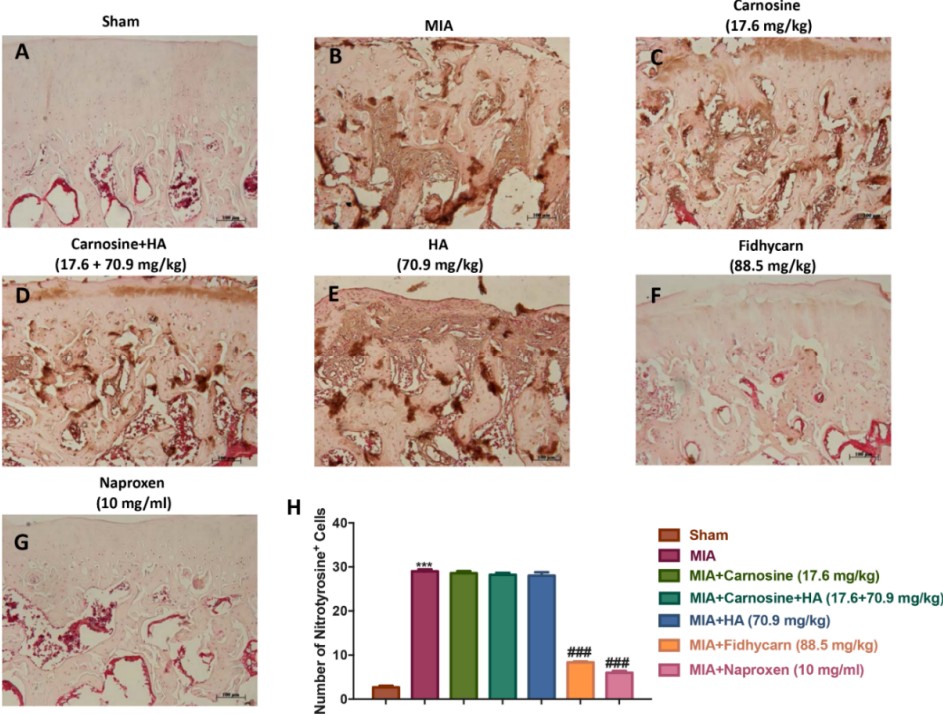

**Figure 8.** The effects of FidHycarn on nitrotyrosine expression. Immunohistochemistry for nitrotyrosine in joint tissues: (**A**) Sham group, (**B**) MIA group, (**C**) MIA+Carnosine, (**D**) MIA+Carnosine+HA, (**E**) MIA+HA, (**F**) MIA+FidHycarn treatment group, (**G**) MIA+Naproxen. The results are expressed as a % of positive pixels (**H**). Figures are representative of at least three independent experiments. *** $p < 0.001$ versus Sham; ### $p < 0.001$ versus MIA.

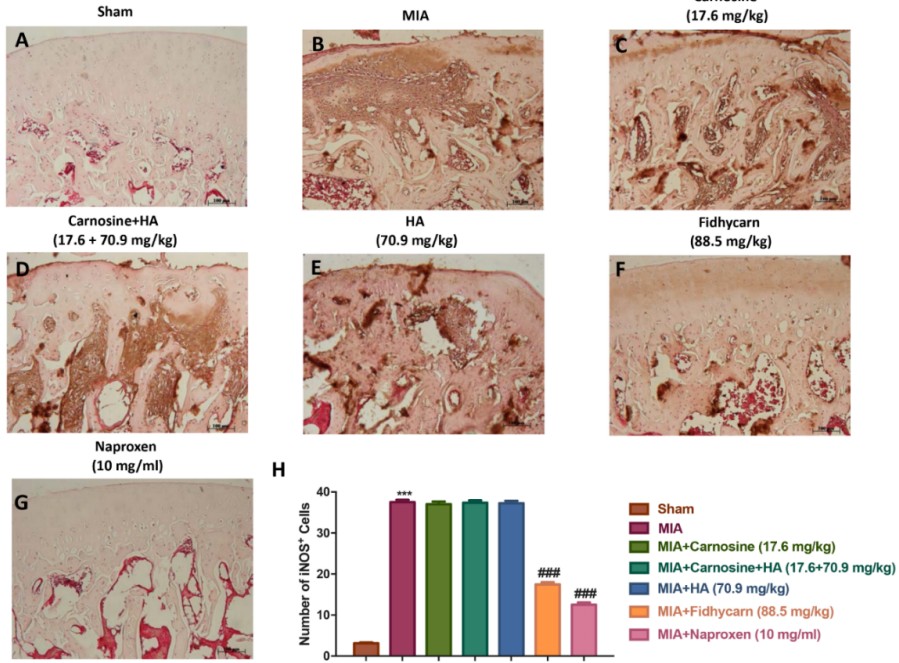

**Figure 9.** The effects of FidHycarn on iNOS expression. Immunohistochemistry for iNOS in joint tissues: (**A**) Sham group, (**B**) MIA group, (**C**) MIA+Carnosine, (**D**) MIA+Carnosine+HA, (**E**) MIA+HA, (**F**) MIA+FidHycarn treatment group, (**G**) MIA+Naproxen. The results are expressed as a % of positive pixels (**H**). Figures are representative of at minimum three independent experiments. *** $p < 0.001$ versus Sham; ### $p < 0.001$ versus MIA.

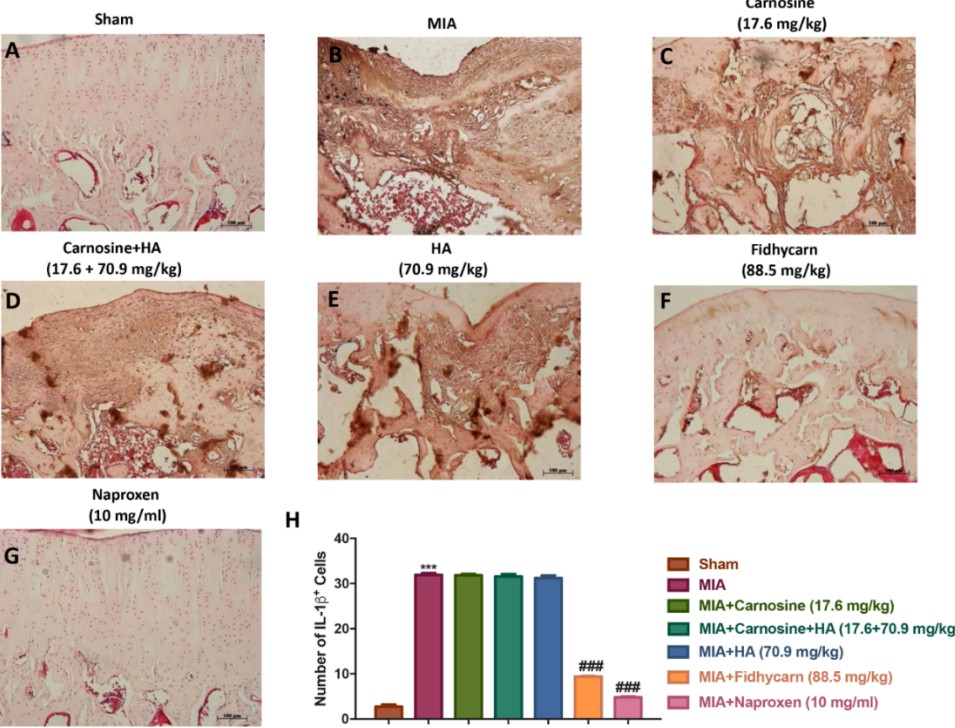

**Figure 10.** The effects of FidHycarn on IL-1 expression. Immunohistochemistry for IL-1β in the joint tissues: (**A**) Sham group, (**B**) MIA group, (**C**) MIA+Carnosine, (**D**) MIA+Carnosine+HA, (**E**) MIA+HA, (**F**) MIA+FidHycarn treatment group, (**G**) MIA+Naproxen. The results are expressed as a % of positive pixels (**H**). Figures are representative of at least three independent experiments. *** $p < 0.001$ versus Sham; ### $p < 0.001$ versus MIA.

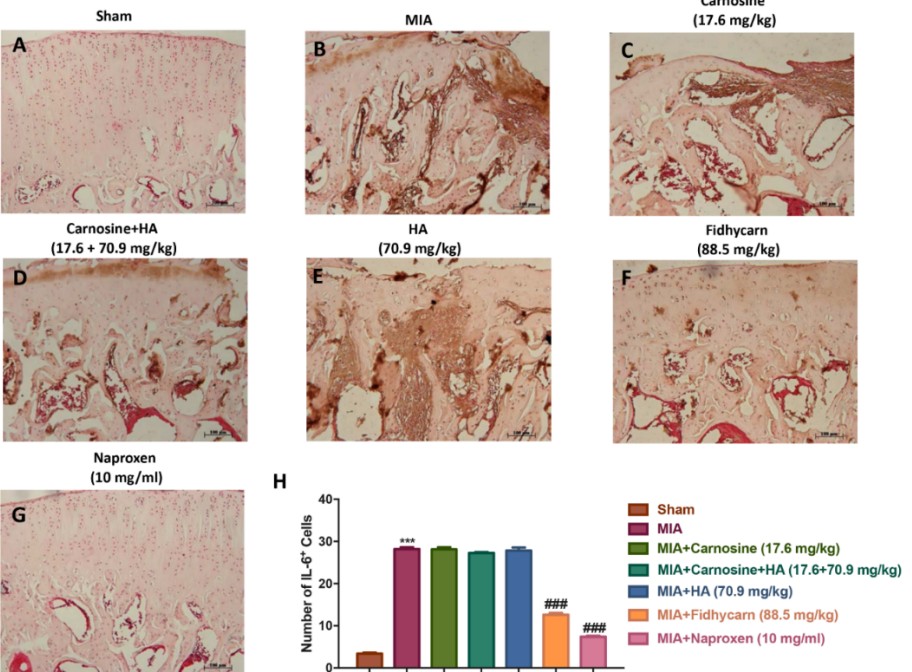

**Figure 11.** The effects of FidHycarn on IL-6 expression. Immunohistochemistry for IL-6 in the joint tissues: (**A**) Sham group, (**B**) MIA group, (**C**) MIA+Carnosine, (**D**) MIA+Carnosine+HA, (**E**) MIA+HA, (**F**) MIA+FidHycarn treatment group, (**G**) MIA+Naproxen. The results are expressed as a % of positive pixels (**H**). Figures are representative of at minimum three independent experiments. *** $p < 0.001$ versus Sham; ### $p < 0.001$ versus MIA.

## 7. Discussion

OA is a very common degenerative disease. This disease is characterized by structural loss and functional impairment of the joints, leading to disability [24–26]. Whereas there is no cure for OA to date, most of the treatments available are intended to control pain and uphold joint function. Usually, the treatment of OA begins with safer and less invasive therapies, such as physical activity. This is because the benefits of exercise can lead to the prevention of obesity and maintenance of normal joint health [27,28]. If OA pain cannot be controlled by physical therapy, different pharmacological treatments are available. Paracetamol is usually the first line drug therapy in OA [29,30]. If paracetamol is not successful, the next level of pharmacological treatment varies according to the patient, but usually involves the use of non-steroidal topical or oral anti-inflammatory drugs (NSAIDs); topical capsaicin; and injections of intra-articular corticosteroids and opioids [31,32]. However, these drugs expose patients to the risk of liver and kidney damage [30]. In addition, these medicines often offer only partial relief [33,34] and do not help the body rebuild the damaged joint cartilage [35]. Numerous studies have shown a role of ROS in the pathogenesis of chronic inflammatory arthropathy, such as OA [36]. Therefore, researchers are considering a multifactorial approach to managing OA.

HA is secreted by chondrocytes and used for cartilage synthesis. According to the research, HA is helpful in the control of OA as it interferes with pain mediators and crucial enzymes (such as metalloproteinase) that are responsible for the digestion and destruction of healthy cartilage tissue [37,38]. HA is generally administered intraarticularly, and also orally [39–41]. New products are frequently being developed that modify the composition of the molecule and associate it with other drugs to maximize the effect [37,42].

CARN is a dipeptide consisting of β-alanine and L-histidine. The anti-inflammatory potential of CARN has been studied in autoimmune diseases [43]. Furthermore, the ability of CARN as an antioxidant, and the anti-glycating and toxic metal-ion chelating properties are well documented [44]. In previous studies the capacity of CARN in the regression of the senescence of cultured human fibroblasts and in delaying aging has been highlighted; however, the mechanisms remain uncertain [45]. Furthermore, it has been shown that CARN reduces the oxidative damage on chondrocytes and prevents cartilage degradation in the "arthritic" joints [43]. However, the anti-inflammatory potential of CARN in OA is still poorly studied. In a recent report, the potential protective effect of an oral CARN supplement was assessed to improve type 2 diabetes-induced OA [46].

Based on the studies performed on HA and CARN, the aim of this study was to evaluate the synergistic effect of a new formulation called FidHycarn, consisting of the covalent conjugation of these two substances, in an in vivo MIA model. In this regard, we have shown that the oral administration of the conjugated FidHycarn was able to improve behavioral deficits and to reduce macroscopic, histological, and radiographic alterations in a more significant way compared to the administration of single compounds or carnosine+HA association.

Oxidative stress shows a potential role on the pathogenesis of OA. In subjects suffering from OA there is a reduction in antioxidants [47]. Proinflammatory cytokines (IL-1β, IL-6, and TNF-α), chemokines (MIP-Iα and MIP-2), and NO also play an important function in the pathogenesis of joint damage [48–50]. In this regard, in this study, it was found that in rats with OA, treated with carnosine or HA or carnosine+HA, the levels of pro-inflammatory cytokines and chemokines were not significantly reduced. In contrast, in FidHycarn-treated rats, a significant diminution of these proteins was observed. The same trend was highlighted when the expression of nitrotyrosine and iNOS was evaluated. This study has demonstrated that FidHycarn at dose of 88.5 mg/kg was able to reduce the joint inflammation and cartilage degeneration induced by MIA i.ar. in a more significant way compared to the of HA and/or CARN treatments. The effects of FidHycarn were only in some cases statistically different to Naproxen (used as positive control).

The greater efficacy of FidHycarn is due to the ability of hyaluronic acid to protect carnosine from enzymatic degradation. In fact, the presence of HA in the FidHycarn conjugate allows for a greater resistance to the action of serum carnosinases, guaranteeing a remarkable stability and consequently a

higher activity than for unconjugated carnosine [51,52]. Therefore, these two molecules, when they work in synergy, exhibit greater biological activity and enhanced antioxidant activity.

Furthermore this study has demonstrated that FidHycarn using two safe products: carnosine and HA has a bio-pharmacological effect comparable to the treatment with naproxen (a standard treatment for OA but with proven side effects) [53], at the clinical dose per kg, for almost all the parameters assessed, without significant statistical differences. In conclusion, the results obtained demonstrate for the first time in an animal model of osteoarthritis the effectiveness of oral administration of the conjugate FidHycarn compared to the administration of the individual components. Therefore, the new FidHycarn could represent an interesting therapeutic strategy to combat osteoarthritis.

**Author Contributions:** Conceptualization, R.D.P., A.S., and S.C.; methodology, R.S., D.I., A.F.P., E.G., and R.F.; validation, S.C., R.S., D.I. and R.D.P.; formal analysis, R.S., M.C., E.R., S.V., M.P., V.G., S.S., and L.M.; investigation, R.S., D.I., M.C., and R.D.; data curation, R.D.P., D.I., S.C., and A.S.; writing—original draft preparation, R.S.; writing—review & editing, D.I. and A.S.; visualization, R.C.; supervision, R.D.P., A.S., and S.C. All authors have read and agreed to the published version of the manuscript.

**Funding:** This research received no external funding.

**Acknowledgments:** The authors would like to thank Valentina Malvagni for editorial support with the manuscript.

**Conflicts of Interest:** Fidia Farmaceutici S.p.A. employees (Antonella Schiavinato, Susanna Vaccaro, Mariafiorenza Pulicetta, Luciano Messina) and the other authors declare that there are no conflict of interest.

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
