# Peer review of "The Protective Effect of New Carnosine-Hyaluronic Acid Conjugate on the Inflammation and Cartilage Degradation in the Experimental Model of Osteoarthritis"

_applsci, doi:10.3390/app10041324_

Round 1

Reviewer 1 Report

In this manuscript, the experiments were well performed, and the results are consistent: thus, the manuscript is interesting for the readers and acceptable for the journal.

Author Response

Thank you for your opinion on the manuscript

Reviewer 2 Report

The authors should consider the followings:

the authors should give details rationale for why Carnosine or HA alone showed no or limited potency, while all of a sudden the formulation demonstrated the significant effect. Is DMSO being removed from the formulation and the % used? Given that, DMSO may be an active anti-inflammatory agent. please provide the data for the purity of FidHycarn. And please provide the list of ingredient(s) that is of over 1% of the total composition Were there any blinding of the groups, in the process of scoring (and or counting/ analysis) of the results in Figure 3, Figure 4 Figure 6, Figure 7 to 9? Since oral treatment is used, did the authors assess the general toxicity of the drugs used? If yes, please provide the evidences. i.e. liver toxicity AST, ALT readings etc. on the 4th and 5th affiliation (seems to be a private company or entity), please declare clearly whether there is any conflict of interest (financial) to the article. Please state clearly the novelty of this research in your abstract and conclusion. please give rationales of why the treatments were only started from Day 3 after MIA induction. Were the staining processed in a evenly distributed groups of samples? Please briefly describe the process of verification of drugs provided from FIDIA. For traceability purposes, please also provide the cat.no. (and/or clone number of the antibodies used) in the experiments. Proposed conjugation of HA/ Carnosine in the formulation may be illustrated in graphical figure in the article. Methods of elucidation of such conjugation should be provided. The authors should highly consider using English proof-reading services by language professional.

Reviewer 3 Report

Minor:

Manuscript needs to be edited for English. There are several typo errors.

Major:

FidHycarn yield better results than Carnosine+HA combination. Although, authors presented their view on this matter. However, a more detailed justification is needed for this claim. Why the synergistic effect is only vivid following covalent interaction? What are the systemic side effects of FidHycarn?
